# A solution-processable and ultra-permeable conjugated microporous thermoset for selective hydrogen separation

Wei Liu [1,2,6], Shu-Dong Jiang [2,6], Youguo Yan [3,6], Wensen Wang [3], Jing Li [1], Kai Leng [1], Susilo Japip [2], Jiangtao Liu [2], Hai Xu [1], Yanpeng Liu [1], In-Hyeok Park [1], Yang Bao [1], Wei Yu [1], Michael D. Guiver [4,5✉], Sui Zhang [2✉] & Kian Ping Loh [1✉]

The synthesis of a polymer that combines the processability of plastics with the extreme rigidity of cross-linked organic networks is highly attractive for molecular sieving applications. However, cross-linked networks are typically insoluble or infusible, preventing them from being processed as plastics. Here, we report a solution-processable conjugated microporous thermoset with permanent pores of ~0.4 nm, prepared by a simple heating process. When employed as a two-dimensional molecular sieving membrane for hydrogen separation, the membrane exhibits ultrahigh permeability with good selectivity for $H_2$ over $CO_2$, $O_2$, $N_2$, $CH_4$, $C_3H_6$ and $C_3H_8$. The combined processability, structural rigidity and easy feasibility make this polymeric membrane promising for large-scale hydrogen separations of commercial and environmental relevance.

[1] Department of Chemistry, National University of Singapore, 3 Science Drive 3, 117543 Singapore, Singapore. [2] Department of Chemical and Biomolecular Engineering, National University of Singapore, 4 Engineering Drive 4, 117585 Singapore, Singapore. [3] School of Materials Science and Engineering, China University of Petroleum, 266580 Qingdao, Shandong, China. [4] State Key Laboratory of Engines, Tianjin University, 300072 Tianjin, China. [5] Collaborative Innovation Center of Chemical Science and Engineering (Tianjin), 300072 Tianjin, China. [6] These authors contributed equally: Wei Liu, Shu-Dong Jiang, Youguo Yan. ✉email: guiver@tju.edu.cn; chezhangsui@nus.edu.sg; chmlohkp@nus.edu.sg

Hydrogen purification from the steam-reforming of methane, and recovery from offgas streams in a refinery or other industrial processes present a good opportunity to boost the production of clean energy[1]. Membrane-based gas separation provides an energy-efficient solution to selective hydrogen separation, but performance is often limited by the trade-off between permeability and selectivity[2–5]. In the past decade, advances have been made with microporous solids possessing extremely rigid network structures and well-defined pores, such as zeolites and metal organic frameworks, from which membranes with thin permselective layers have high permeability and good selectivity[6–9]. However, industrial membranes are dominated by polymers that are fabricated by simple solution-processing techniques[10]. Currently, commercial gas separation membranes are fabricated from a few polymers with relatively low permeability and high selectivity[11]. The low gas permeability of commercial membranes requires large membrane size for sufficient production, which is a critical cost challenge for industrial applications. Thus, microporous polymers that possess rigid network structures, high gas permeability, good selectivity, and solution-processability are highly desirable for next-generation gas separation membranes.

Among microporous organic materials, porous organic cages (POCs)[10,12,13] and polymers of intrinsic microporosity (PIMs)[11,14–17] are extensively studied on account of their microporosity and good solution-processability. POCs are shape-persistent organic molecules with accessible intrinsic cavities, which can be aligned during slow solvent evaporation, leading to homogenous microporous membranes assembled via van der Waals (vdWs) interactions. Cooper and co-workers prepared POC membranes for selective hydrogen separation, in which a cage molecule CC3 with a window diameter of ~0.6 nm achieved a $H_2$ permeability of 226 barrer and a $H_2/N_2$ selectivity of 30[10]. Microporosity in PIMs arises from loose packing of contorted and rigid macromolecular chains, and large area membranes with moderately good mechanical strength can be readily fabricated by solution processing. However, cage molecules and linear polymer chains assembled by weak vdW forces can easily slide over one another when thermally or mechanically agitated, resulting in limited selectivity[14,18]. In addition, $CO_2$ has a substantial plasticization effect on the membranes at high temperature, leading to further performance decay[18]. Therefore, there is a need to develop materials that possess chemically bonded networks for high performance gas separation, while simultaneously being solution-processable.

Conjugated microporous polymers (CMPs) are π-conjugated networks interconnected by irreversible aryl-aryl covalent bonds. Thus, the microporosity in CMPs is persistent in the presence of water vapor, acid gases and hydrocarbons, even at elevated pressure and temperature[19–22]. However, CMPs have extremely rigid cross-linked structures, which inevitably lead to poor processability. Developing simple and scalable processing routes is of paramount importance for their practical application[23,24]. In addition, CMPs normally possess a broad pore size distribution (10–30 Å), which are not suitable for gas separation. To simultaneously meet the demands of structural rigidity and solution-processability, we have developed a solution-processable conjugated microporous thermoset (CMT) with intrinsic pores of ~0.4 nm and a specific surface area of ~840 $m^2 g^{-1}$, based on a simple thermosetting process. We show that the polymeric networks based on aryl-aryl bonds can be prepared into various shapes and morphologies, driven by the substrate onto which the CMT is deposited. More importantly, 2D CMT nanosheets are solution-processed into large-size molecular sieve membranes. With contributions from both intrinsic microporosity and interlayer spacing, CMT membranes achieve a $H_2$ permeability as high as 28280 barrer and good selective $H_2$ separation over $CO_2$, $O_2$, $N_2$, $CH_4$, $C_3H_6$, and $C_3H_8$. In addition, CMT membrane shows no performance decay up to 700 h at 150 °C, demonstrating its remarkable stability and persistence of microporosity.

## Results

**Thermosetting polymerization.** The precursor 3,6,12,15-Tetra-bromotetrabenzo[a,c,h,j]phenazine (3-TBTBP, $M_w$ 696) was synthesized by a condensation reaction between 3,6-dibromo-phenanthrene-9,10-diaminium chloride (3,6-DBPDA) and 3,6-dibromophenanthrene-9,10-dione. The tetrabrominated planar molecule is insoluble in common organic solvents due to strong intermolecular π–π stacking. To perform structural characterization, single crystals were prepared by sublimation of the prepared crude material in a tube furnace. The structure information of 3-TBTBP crystals is provided in Supplementary Fig. 1. Our studies clearly show that 3-TBTBP crystals sequentially undergo sublimation, melting, and polymerization during heating. Thermal gravimetric analysis (TGA) reveals that there are two weight loss stages for 3-TBTBP in the range of 200–900 °C in $N_2$ atmosphere (Fig. 1b). The first stage starting at ~450 °C is attributed to sublimation. The second stage is caused by debromination of 3-TBTBP, which begins at ~520 °C. To further understand the phase change during the heating process, the precursor powder is enclosed in an Al crucible for differential scanning calorimetry (DSC) analysis. The DSC curve was obtained in the range of 25–600 °C in $N_2$ atmosphere (Fig. 1c). It shows no obvious peak before 500 °C, indicating that the sublimation of precursor crystal is largely suppressed in the closed Al crucible. The endothermic peak at 509 °C originates from the melting of the precursor. The following broad exothermic peak starts at ~515 °C and can be ascribed to exothermic debromination and concomitant polymerization[25,26]. This exothermic feature is consistent with solid-state polymerization of halogenated thiophene derivatives[25]. This broad peak, which can be deconvoluted into three peaks at ~535 °C, ~550 °C, and ~562 °C, reflects that the cleavage of the four C–Br bonds in 3-TBTBP have different activation energies. The DSC results indicate that before polymerization, 3-TBTBP crystals melt at ~509 °C to form a molecular liquid. The unique thermal properties of 3-TBTBP allow the synthesis of a conjugated microporous thermoset.

Our thermosetting polymerization experiments were conducted in a tube furnace with an Ar gas flow of 100 ml $min^{-1}$. In the tube furnace, samples were put in a covered jar to minimize the disturbance of gas flow. The sample zone was heated to 540 °C at a heating rate of 15 °C $min^{-1}$ and held at 540 °C for 2 h to produce the CMT. X-ray photoelectron spectroscopy (XPS) spectra (Supplementary Fig. 2) show that Br peaks (Br $3p_{3/2}$ at 190.5 eV and Br $3p_{1/2}$ at 183.9 eV) disappear after polymerization, in agreement with the FTIR measurements (Supplementary Fig. 3). FTIR spectra show that the absorption band corresponding to aryl-bromine vibration mode at 548 $cm^{-1}$ largely disappears, while the characteristic bands of phenazine linkages at 1361 and 1400 $cm^{-1}$ remain after polymerization. Elemental analysis shows that only 4.38 wt% of Br remains in the polymer product (Supplementary Table 1), indicating a high degree of polymerization.

**Processability.** To demonstrate its processability, CMT was prepared in 3D, 2D, and 1D forms, depending upon the substrate (details are provided in Methods). Figure 1a shows the schematic illustration of the thermosetting process. Needle-like 3-TBTBP crystals were sublimed and/or melted, so that the precursor vapor and/or liquid were readily shaped into objects with desirable

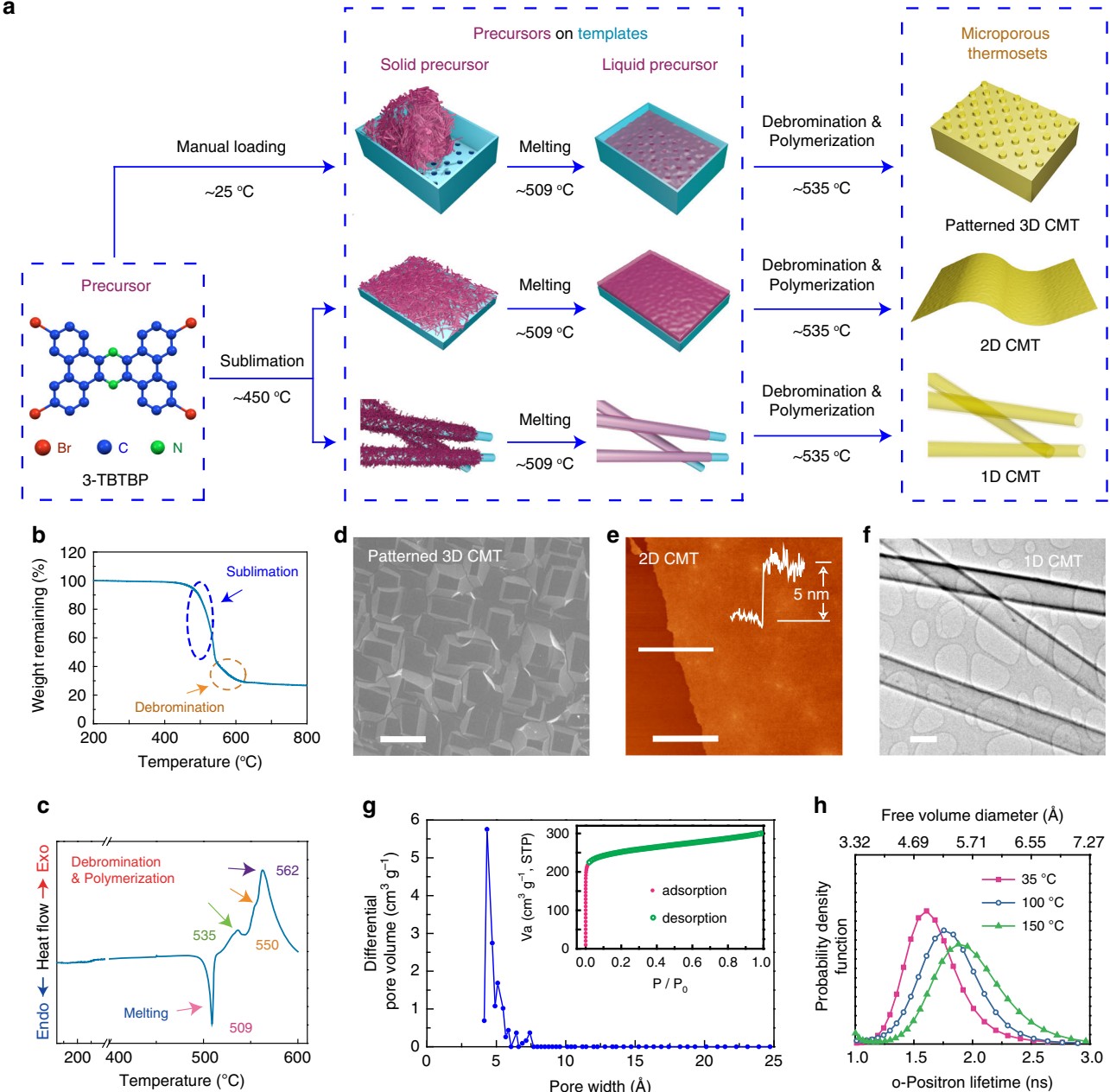

**Fig. 1 Synthesis and characterization of conjugated microporous thermoset. a** Schematic illustration of the thermosetting process to make 3D, 2D, and 1D CMT, depending upon the substrate (a holder with patterned bottom plate for patterned 3D CMT; Silicon wafer for 2D CMT; Cu nanowires for 1D CMT). **b** Thermal gravimetric analysis of 3-TBTBP. **c** Differential scanning calorimetry curve of 3-TBTBP. **d** SEM image showing the surface of a patterned 3D CMT. **e** AFM topography image of ~5-nm-thick CMT film on Si wafer; Insert: corresponding height profile. **f** TEM image of 1D CMT nanotubes. **g** Pore size distribution derived based on Ar adsorption data; Inset: Ar isotherm profile of the CMT measured at 86 K. **h** The o-positron lifetime (and hence free volume diameter) distribution in CMT at 35 °C, 100 °C, and 150 °C based on PALS measurements. The scale bars are 20 μm, 3 μm, and 200 nm for **d**, **e**, and **f**, respectively.

morphologies using suitable substrates, which were further cured via thermal-initiated debromination and concomitant C–C cross-coupling reaction. Patterned 3D CMT was prepared by melting a large amount of precursor crystals in a patterned mold followed by curing via polymerization. Figure 1d shows SEM image of the patterned surface of 3D CMT, which has a homogeneous and dense morphology without any inner voids (Supplementary Fig. 6).

The sublimation of 3-TBTBP produces precursor vapor (as demonstrated in TGA), which adsorbs readily on any exposed surfaces. This was exploited to prepare 2D CMT films and

nanotubes by vapor phase deposition, as demonstrated in Fig. 1a. Figure 1e shows an AFM topography image of a CMT film homogeneously polymerized on Si wafer. The corresponding height profile (insert) shows that the film has a thickness of ~5.0 nm. The film fully covers the Si wafer and its thickness can be easily tuned by the ratio of precursor to available substrate template surface area (Supplementary Fig. 7). CMT nanotubes were produced by polymerizing 3-TBTBP on Cu nanowires followed by etching away the Cu nanowires with HCl aqueous solution. Figure 1f shows CMT nanotubes under TEM. The crack-free CMT films and nanotubes suggest that precursor vapor

adsorbs uniformly on the substrate surfaces. Subsequently, the adsorbed precursor layer melts to form a smooth and continuous film, which is further cured to produce the cross-linked networks.

**Porous structure**. The porosity of CMT has been cross-verified by Ar adsorption/desorption isotherms and positron annihilation lifetime spectroscopy (PALS). As shown in Fig. 1g, CMT shows a type 1 isotherm with a steep uptake at $P/P_0$ below 0.01, which is a typical characteristic for microporous materials. The Brunauer–Emmett–Teller (BET) surface area and pore volume are evaluated to be 840 $m^2 g^{-1}$ and 0.39 $cm^3 g^{-1}$, respectively. The adsorption isotherm fitted with non-local density functional theory model reveals that the pore-size-distribution for CMT is mainly at 0.4–0.5 nm, which agrees well with the positron annihilation lifetime spectrum (PALS) analysis. As shown in Fig. 1h, the free volume diameter of CMT as measured by PALS at 35 °C has a maximum at 4.89 Å, consistent with Ar sorption measurements. With an increase in temperature, the size of accessible voids is slightly broadened due to the more active thermal motion of the building units. A sub-nanometer pore size and remarkably narrow pore size distribution of CMT are uncommon, especially when compared to conjugated microporous polymers synthesized in solvents with noble metal catalysts under

stirred conditions[19,22,27]. In this case, the endogenous liquid-state polymerization (that is, without the presence of solvents, initiators, or catalysts) is able to achieve a homogeneous porous structure, because precursor molecules are polymerized in the isotropic liquid state, without randomization effects of solvents and catalysts. More importantly, the endogenous polymerization method allows the potential scale-up production of CMT due to its low cost, high yield (close to 100%) and easy processing.

**Membrane fabrication**. Interfacial polymerization strategies have previously been applied to produce films composed of cross-linked networks[28–32]. In addition to their large pore size distribution[28,29], a further disadvantage of the polymerization process is the need for a highly functional substrate[28,32]. In contrast, our CMT can be prepared as uniform ultrathin sheets with controllable thickness on any arbitrary substrates. As shown in Fig. 2b, CMT is polymerized uniformly on the surface of NaCl crystal substrate; the latter can be removed by soaking in water, and the polymeric sheets are recovered for use. Most importantly, the obtained CMT ultrathin sheets are highly dispersible in common organic solvents (Fig. 2d), thus allowing them to be processed into large-area membranes with controllable thickness by simple filtration. Figure 2c shows a CMT/CHCl₃ dispersion

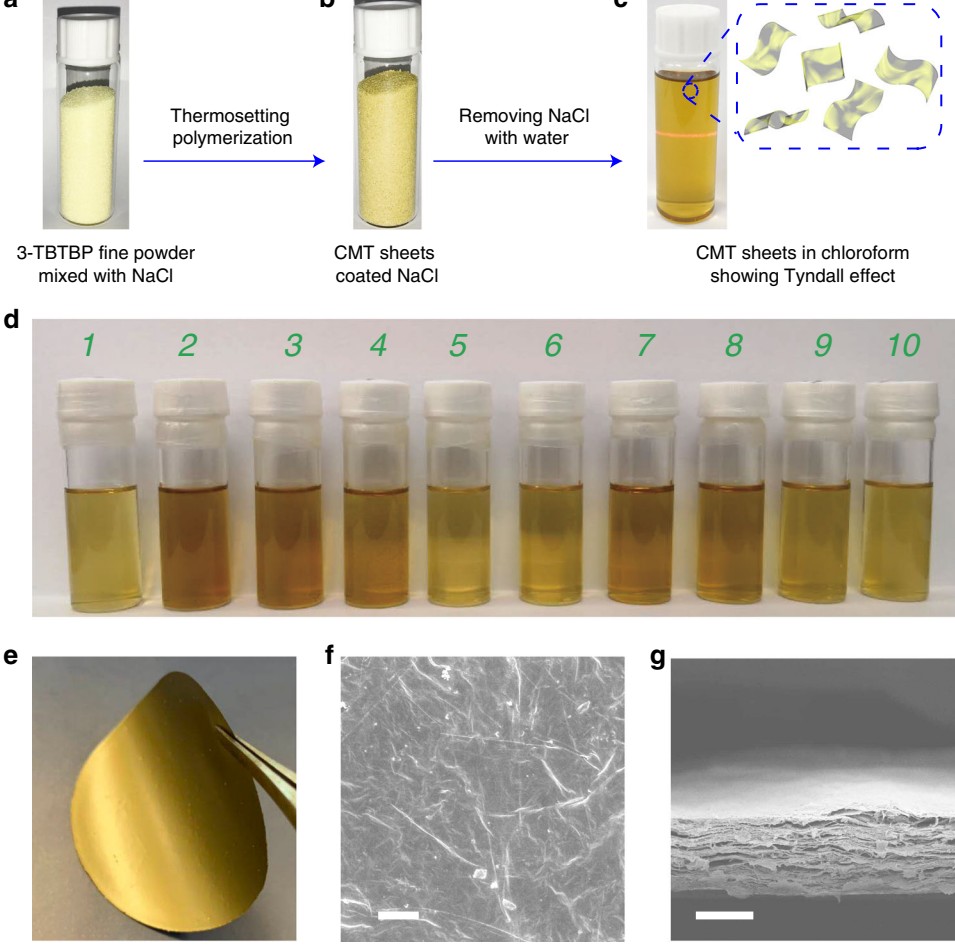

**Fig. 2 Synthesis and characterization of CMT membrane. a** An optical image of a mixture of 3-TBTBP precursor fine powder and NaCl crystals. **b** An optical image of CMT coated NaCl. **c** CMT/Chloroform dispersion showing a Tyndall effect. **d** CMT dispersions (concentration: 0.05 mg ml⁻¹); solvent: (1) dichloromethane, (2) ethanol, (3) methanol, (4) hexane, (5) diethyl ether, (6) acetone, (7) dimethylformamide, (8) dimethyl sulfoxide, (9) tetrahydrofuran, and (10) isopropanol. **e** A free-standing CMT membrane with a diameter of ~47 mm prepared by filtration. **f** A SEM image showing the surface of a CMT membrane. **g** A SEM image showing the cross-section of a free-standing CMT membrane. The scale bars are 2 μm and 10 μm for **f** and **g**, respectively.

displaying an obvious Tyndall effect; it remains stable even after storing under ambient conditions for two weeks, with no sign of precipitation. The intrinsic micropores in CMT, rimmed by N groups, provide interfaces for solute/solvent interactions and improve wettability. The stability of the dispersion should be contrasted with the strong tendency of exfoliated graphene or molybdenum disulfide sheets to restack and precipitate[33,34]. Fig. 2e is a photo of a free-standing CMT membrane prepared by filtration of CMT dispersion, which shows a shiny golden color. The yellow CMT nanosheets dispersion shows a broader UV-vis absorption than the precursor 3-TBTBP (Supplementary Fig. 11a). The Tauc plot extracted from the UV-vis spectrum reveals a band gap of ~2.45 eV (Supplementary Fig. 11b), which matches well with the value derived from the photoluminescence (PL) spectrum of CMT film on Si wafer, showing a peak at ~525 nm (~2.36 eV) (Supplementary Fig. 12). The optical data suggests the semiconductive nature of conjugated CMTs. The cross-section of a CMT membrane observed by SEM is presented in Fig. 2g, revealing its lamellar-like structure. The plan view shows

a crack-free but slightly wrinkled surface (Fig. 2f). The thickness of each sheet can be easily tuned by the weight ratio between 3-TBTBP and NaCl. For example, when 1 mg of 3-TBTBP is homogeneously mixed with 6 g of NaCl and polymerized, the thickness of the sheets is around 5 nm. Due to the templating effect by NaCl, the size of the CMT sheets is defined by the area of the salt crystal surfaces and we can easily obtain sheets as large as 100 μm after the recovery process (Supplementary Fig. 13). The TEM image of CMT sheets shows a homogeneous morphology (Supplementary Fig. 14).

**Gas separation performance**. Gas permeation through the CMT membranes was measured in a permeation cell at 30 °C and a transmembrane pressure of 1 bar. Figure 3a shows that the permeability of various gases scales inversely with their kinetic diameters. A 1-μm-thick CMT membrane exhibits ultrahigh permeabilities for He (24200 barrer) and $H_2$ (28280 barrer), while relatively much lower permeabilities are observed for $CO_2$ (4480

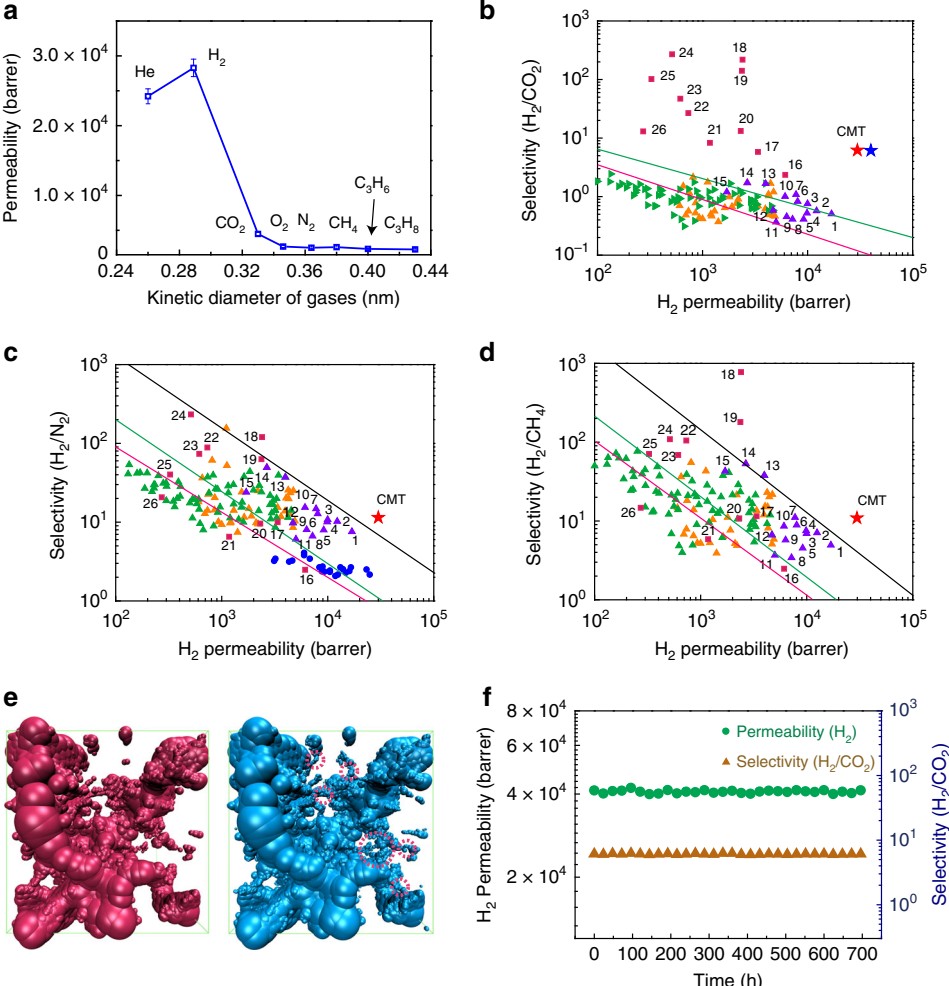

**Fig. 3 Gas separation performance of CMT membranes. a** Single-gas permeabilities (He, $H_2$, $CO_2$, $O_2$, $N_2$, $CH_4$, $C_3H_6$, and $C_3H_8$) through a 1-μm-thick CMT membrane at 30 °C and a transmembrane pressure of 1 bar. **b–d** Robeson plots for $H_2/CO_2$, $H_2/N_2$, and $H_2/CH_4$ gas pairs with the 1991 upper bounds indicated by a pink line, 2008 upper bounds by a green line and proposed 2015 upper bounds by a black line. The single-gas permeation data of CMT membrane (red star) is compared with accumulated literature data for state-of-the-art ultrapermeable PIMs (violet triangle), microporous solids (zeolites, MOFs, POPs) and inorganic 2D materials (red square), and other polymers (ladder PIMs: orange triangle; PIM-PIs: green triangle; ultrapermeable poly(trimethylsilylpropyne) (PTMSP): blue circle). Information on the data points and their corresponding references are presented in Supplementary Table 2. **e** Accessible void spaces for $CO_2$ (left) and $H_2$ (right) in CMT model; The red circles in the right picture indicate additional accessible void spaces for $H_2$, compared with that of $CO_2$. **f** Long-term test of an equimolar $H_2/CO_2$ mixed-gas mixture through a 1-μm-thick CMT membrane at 150 °C and a transmembrane pressure of 1 bar. The blue star in **b** indicates the permeation data from $H_2/CO_2$ mixed-gas measurement at 150 °C.

barrer), $O_2$ (2680 barrer), $N_2$ (2500 barrer), $CH_4$ (2590 barrer), $C_3H_6$ (2330 barrer), and $C_3H_8$ (2260 barrer). The ideal selectivities of $H_2$ over $CO_2$, $O_2$, $N_2$, $CH_4$, $C_3H_6$, and $C_3H_8$ are ~6.3, ~10.6, ~11.3, ~10.9, ~12.1, and ~12.5, respectively. This demonstrates the molecular sieving effect of CMT membranes. In addition, only slight variations in permeability and selectivity are observed when the membrane thickness is increased from 500 nm to 13 μm (Supplementary Fig. 16), suggesting that the membranes have a low defect density.

To overcome the limit of conventional polymeric membranes, next-generation membranes for hydrogen separation require fast hydrogen permeation with sufficiently high selectivity. This has been done through the design of microporous materials, such as PIMs and MOFs[11,17], or by the creation of additional transport channels via the interlayer spacing between stacked 2D nanosheets, such as graphene oxide and MXene[35,36]. Figure 3b–d compile the Robeson plots for $H_2/CO_2$, $H_2/N_2$, and $H_2/CH_4$ gas pairs, including data for CMT membranes, and state-of-the-art ultrapermeable PIMs, microporous solids with network structures (zeolites, MOFs, porous organic polymers, etc.), inorganic 2D materials (GO, MXenes), as well as other organic polymeric membranes (Supplementary Table 2). The permeability-selectivity data of CMT membranes fall well above the 2008 upper bounds for all gas pairs, and fall above the recently proposed 2015 upper bounds for $H_2/N_2$ and $H_2/CH_4$ gas pairs. In particular, the ultrapermeable nature of CMT membranes towards hydrogen is clearly seen on the three plots. The $H_2$ permeability of CMT membrane, at similar or higher selectivity, is higher than the state-of-the-art ultrapermeable PIMs for all the gas pairs. The porosity of CMT polymers is 840 $m^2\,g^{-1}$ (Fig. 1e), which is comparable to most microporous polymers. To reveal the benefits of layered 2D structure, 3D CMT membranes were prepared by in-situ growth on anodized aluminum oxide (AAO) substrate and the corresponding $H_2$ permeability is 7100 barrer, which is only ~24% of the permeability of 2D CMT membranes (Supplementary Fig. 17). This suggests that the combination of both in-plane microporous structure and inter-layer free spacing between CMT sheets are mainly responsible for the extremely high permeability of 2D CMT membrane.

To determine the mechanism of gas separation in CMT, molecular dynamic simulations were performed. The polymerization was simulated in a periodic cubic cell with dimensions of $50 \times 50 \times 50$ Å$^3$ and containing 160 monomers (7360 atoms) using Polymatic algorithm[37]. This generates a microporous polymeric network that reproduces the experimental mass density, accessible surface area (Ar sorption) and pore size distribution of CMT well (Supplementary Table 3 and Supplementary Fig. 20). We use $H_2$ and $CO_2$ as the probe gas for the CMT model. It is found that CMT has a larger accessible volume for $H_2$ (19.58% of membrane volume) than $CO_2$ (16.28% of membrane volume). This suggests that CMT is able to accommodate more $H_2$ than $CO_2$, leading to a higher concentration gradient for $H_2$, which facilitates transmembrane transport[38]. More importantly, some channels allow $H_2$ transport but block $CO_2$, as shown in Fig. 3e. The average $H_2$ transmembrane path calculated based on accessible void space is ~710 Å, which is much shorter than that of $CO_2$ (~3715 Å), thus contributing largely to the ultrahigh $H_2$ permeability. The CMT membrane was further studied in $H_2/CO_2$ mixed gas separation to demonstrate its long-term stability at elevated temperature. Similar to other polymeric membranes[11], the $H_2$ permeability increases by more than 40% to 40,680 barrer when the temperature increases from 30 °C to 150 °C (Supplementary Fig. 18). This can be explained by the broadened free volume size of CMT at higher temperature (as confirmed by PALS), which facilitates diffusive transport of $H_2$. The optimal $H_2/CO_2$

selectivity is 6.05 at 150 °C. Most importantly, CMT polymers are thermally stable and can potentially be operated up to 500 °C. Both the permeability and selectivity remain constant for 700 h at 150 °C (Fig. 3f).

## Discussion

Overall, compared with cross-linked porous solids with poor processability, CMT is solution processable and possesses interlayer channels that allow fast redistribution of the permeate gas molecules, leading to much higher $H_2$ permeability. The permeability of the CMT is also higher than most state-of-the-art ultrapermeable PIMs, at higher ($H_2/CO_2$) or similar selectivity ($H_2/N_2$ and $H_2/CH_4$)[11,17]. In addition, the rigid structure of CMT due to the conjugated C–C connected scaffolds prevents the highly homogeneous pores from collapsing into isolated voids and allows for inner pore connectivity at elevated temperatures. Therefore, CMT membranes are unique due to the combination of three properties: they offer ultrahigh $H_2$ permeability and solution-processability of PIMs, and yet enjoy stability and enhanced performance at elevated temperatures, typical of cross-linked rigid networks.

In summary, we have successfully synthesized a conjugated microporous thermoset that exhibits the processability of plastics, together with the rigidity and porosity of cross-linked microporous solids. The microporosity is persistent and stable, even at elevated temperature and pressure. CMT has a narrow pore size distribution centered at ~0.4 nm, when combined with its solution processability, large area ultrafiltration membranes of controlled thickness could be fabricated via simple filtration. Due to the coexistence of intrinsic micropores and interlayer spacing between the stacked sheets, CMT membranes exhibit ultrahigh $H_2$ gas permeability with good selectivity and excellent operation at high temperatures. Although we have demonstrated only the gas separation application of CMT here, we anticipate that CMT, as well as its analogs, will also impact other applications, extending to areas such as ionic nanofiltration and water desalination.

## Methods

**Synthesis of 3-TBTBP**. 3,6-dibromophenanthrene-9,10-diaminium chloride (3,6-DBPDA) was synthesized from 3,6-dibromophenanthrene-9,10-dione (3,6-DBPD, purchased from Tokyo Chemical Industry). 3,6-DBPD (2.0 mmol, 0.73 g) and NH$_2$OH•HCl (143.0 mmol, 10.0 g) were added into a mixed solvent of pyridine (6.0 ml) and EtOH (60.0 ml) to make a suspension, which was heated under reflux for 24 h. The formed precipitate 3,6-dibromophenanthrene-9,10-dione dioxime was collected and washed with EtOH to remove pyridinium hydrochloride. The gray precipitate without further purification was dispersed again in 100 ml EtOH, in which a solution of SnCl$_2$ (21.0 mmol, 4.0 g) in concentrated HCl (15 ml) was added. The mixture was stirred at 70 °C for 4 h. After cooling down to room temperature, the precipitate 3,6-DBPDA was collected and washed with water and EtOH several times, followed by drying at 80 °C. To prepare 3-TBTBP, 3,6-DBPDA (1.0 mmol, 0.44 g) and 3,6-DBPD (1.0 mmol, 0.37 g) were suspended in 80 ml acetic acid in a round bottom flask in Ar atmosphere under stirring. The reaction mixture was refluxed at 130 °C for 6 h after adding triethylamine (1.0 ml). The precipitate was collected by filtration, and thoroughly washed sequentially via Soxhlet extraction with ethanol, tetrahydrofuran, and N,N-dimethylformamide. After drying at 120 °C overnight in vacuum oven, 3-TBTBP was obtained (0.65 g, 94%) as yellow powder. Elemental analysis (%): calculated for (C$_{28}$H$_{12}$Br$_4$N$_2$): C, 48.32; H, 1.74; N, 4.02; Br, 45.92; found: C, 47.76, H, 1.97, N, 3.93, Br 45.37. MALDI-TOF mass (negative mode) $m/z$ = 697.3 (calcd. 696.0); No solution NMR spectroscopic data were collected owing to its insolubility. Its structure was confirmed with single crystal X-ray diffraction analysis (CCDC 1909146).

**Single crystal growth**. 3-TBTBP crude powders (~100 mg) were evenly dispersed on a quartz boat, which was inserted into the quartz tube furnace. The quartz tube is connected to a supply of Ar gas and a vacuum pump. The Ar flow was adjusted to 50 ml min$^{-1}$ under vacuum (~35 Pa). The crude powder was heated at the first heating zone at 350 °C to promote sublimation. The second heating zone was set at 320 °C for molecular transport. 3-TBTBP single crystals were deposited at the edge of the third zone which was heated at 260 °C. Single crystals suitable for crystallography analysis can be produced in about 4 h.

**Synthesis of CMT**. The thermosetting polymerization experiments were conducted in a tube furnace with an Ar gas flow of 100 ml min$^{-1}$. In the tube furnace, precursors with various substrates were put in a jar equipped with a lid to minimize the disturbance of gas flow, so that only a tiny amount of precursor 3-TBTBP was evaporated out and deposited at the low temperature area of the tube furnace. Normally, samples were heated to 540 °C at a heating rate of 15 °C min$^{-1}$ and held at 540 °C for 120 min to produce CMT. The synthesis details for CMT monoliths, ultrathin 2D sheets or 1D nanotubes are provided in the Supplementary Information.

**Preparation of CMT membranes**. The CMT polymer membranes were prepared by pressure-assisted filtration in a dead-end filtration cell through polypropylene (100 nm pore size, 47 mm in diameter, Sterlitech), AAO disks (100 nm pore size, 47 mm in diameter, Whatman), or nylon (200 nm pore size, 47 mm in diameter, Whatman) membrane filters under 1 bar. The thickness of the membranes was controlled by volume and concentration of the dispersion. In a typical experiment, 20 ml of 0.025 mg ml$^{-1}$ CMT polymer dispersion in IPA led to a membrane thickness of about 500 nm. All membranes were dried at room temperature under vacuum for 24 h before use.

## Data availability

Research data supporting this publication is available at https://doi.org/10.5281/zenodo.3698962. The source data underlying Figs. 1b–h, 2f–g, 3a, and Supplementary Fig. 16 are provided as a Source Data file. The X-ray crystallographic coordinates for structures reported in this study have been deposited at the Cambridge Crystallographic Data Centre (CCDC), under deposition number CCDC-1909146. The data can be obtained free of charge from The Cambridge Crystallographic Data Centre via www.ccdc.cam.ac.uk/data_request/cif.

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

## Acknowledgements

K.P.L. acknowledges NRF-CRP grant "Two Dimensional Covalent Organic Framework: Synthesis and Applications". Grant number NRF-CRP16-2015-02, funded by National Research Foundation, Prime Minister's Office, Singapore. S.Z. and K.P.L. acknowledge NUS Cross-Faculty Resarch Grant '2D Polymers for Membrane-based Molecular Separation' with grant number R-279-000-582-133, funded by National University of Singapore. Y.Y. acknowledges Shandong Provincial Natural Science Foundation, China (ZR2019MB023). We acknowledge Prof. Neal Tai-Shung Chung for sharing the use of PALS and gas separation equipment, and thank W. Fu, Z.X. Chen, S.H. Choi, C.H. Yao, H. Xiao, and H. Yan for helpful discussions.

## Author contributions

K.P.L. and S.Z. supervised the project. W.L. conceived the ideal of CMT and synthesized the materials. D.S.J, S.J, J.T.L., and S.Z. conducted the gas separation experiment. Y.G.Y. and W.S.W. performed the theoretical calculations. J.L., K.L., and Y.P.L. helped with Raman and PL. I.H.P helped with the crystal file analysis. Y.B. and H.X. helped with AFM. W.Y. helped to collect XPS spectra. W.L., S.Z., M.D.G., and K.P.L analyzed the data and wrote the manuscript with contributions from all the authors.

## Competing interests

The authors declare no competing interests.
