## [Peer Review File · Nature Communications]

Reviewers' comments:

Reviewer #1 (Remarks to the Author):

This is an interesting and potentially useful method of obtaining a highly size selective membrane for gas permeability. I recommend publication with the following amendments.

1. Permeability measurements for helium should be added as this will help to confirm the size selectivity of the membranes.
2. Data should be added for PIM-BTrip (E&ES, 2019, 12, 2733) to the Robeson plots and S table.

Otherwise I am happy to recommend publication of this exciting research.

Reviewer #2 (Remarks to the Author):

The authors describe a thermal polymerization technique to produce a conjugated microporous thermoset (CMT) as gas separation membrane. The authors describe the direct synthesis on several different substrates (Alumina and Polymers, e.g. Nylon) and with different deposition techniques such as filtration. The membrane exhibits extremely high permeabilities up to $\sim 40,000$ barrer (H_2) and moderate selectivity. The presented material is very interesting. However, the manuscript structure is a bit confusing and I see several points necessary to make the paper suitable for publication in Nat. Commun.

Some minor remarks:

Line 159 – “1D nanotubes” – since you have multilayers (thickness of at least 5nm) of your material your nanotubes correspond to 3D structure. I am not convinced that it is only a monolayer. I would simply refer to “nanotubes”, without dimensions.

Line 196 – “concentration” should be concentration

Line 281 – “selectivity” should be selectivity

You should read the paper carefully again to ensure there are no typos anymore.

Some major remarks:

1) Page 8, 171 – you have such a narrow size distribution, that you should be able to see it on the XRD, since it must be ordered in some way. XRD data should be added to your general characterization methods? Compare for PIMs: <https://doi.org/10.1002/adsu.201800044>

2) In Figure 1e there is a nice AFM picture of the single sheet. In Figure 2h there is another AFM image. The latter is not really needed to state your points, everyone would agree that your sheet thickness stays the same after filtration.

3) In Figure 2e you show your membrane prepared by filtration, with the SEM top view and cross section in Figure 2 f/g. This membrane is 20 μm thick and was not used for any of your experiments, as far as I see it. Why do you not give the actual membrane, that was used for your experiments in the main section (which now is in SI Fig. 16). I cannot really understand your filtration technique to prepare a membrane, when you afterwards measure a different membrane, which is directly synthesized.

4) It is completely unclear, where your data originates from. You write that you use a 1 μm thick membrane for H_2/CO_2 mixed gas in line 252-253. Then, in line 257-259 you speak of a 0.5 μm thick membrane, where you collected your single gas permeation data on.

Keep it consistent and measure it all for the same sample (and add error bars from your control experiments). And if you measure it again, make sure to measure everything at elevated temperatures and in mixed gas permeation (see point 5).

5) The next thing I find is Figure 3 – You compare the Robeson plot with your ideal permeability and selectivity - this is not consistent. You should easily use your “real” permeability data from mixed gas measurements of the binary mixtures. I think you would still be better than all the other points. However, there is often a strong variation between ideal and real permeation data. You should also check back on your data points, whereas the data points originate from real or ideal permeation data and if it is comparable.

6) The same point as in my comment (5) – Line 281-282.

7) Line 310-313. Your data seems to be inconsistent here. When heating up neat materials outgoing from general physical knowledge about adsorption and diffusion, even in zeolites, MOFs, COFs and especially in polymers (solution diffusion), your permeability ALWAYS increases, whereas your selectivity ALWAYS decreases. This is the general trade-off in membrane science. I can also not find the data set on this study in the supporting information, where you measure at different temperatures.

After the manuscript is subjected to a major revision considering the points listed above, it may be suitable for a publication in Nat. Comm.

Date: 10 Feb 2020

RE: Point-by-point response for manuscript NCOMMS-19-39024-T

Reply to Reviewer #1 (Remarks to the Author):

This is an interesting and potentially useful method of obtaining a highly size selective membrane for gas permeability. I recommend publication with the following amendments.

Reply: Thanks a lot for the positive comments. Here we would like to reply to your comments:

Comment: 1. Permeability measurements for helium should be added as this will help to confirm the size selectivity of the membranes.

Reply: Thanks for pointing it out. We have included the He permeability in Figure 3a. It is confirmed that similar to H₂, the permeability of small helium gases is much higher than other molecules with larger kinetic diameter.

Action: He permeability data has been added in Figure 3a.

Comment: 2. Data should be added for PIM-BTrip (E&ES, 2019, 12, 2733) to the Robeson plots and S table.

Otherwise I am happy to recommend publication of this exciting research.

Reply: Thanks.

Action: The PIM-BTrip gas data have been included in Robeson plots Figure 3b, 3c, 3d and supplementary table 2 as membrane No. 2.

Reviewer #2 (Remarks to the Author):

The authors describe a thermal polymerization technique to produce a conjugated microporous thermoset (CMT) as gas separation membrane. The authors describe the direct synthesis on several different substrates (Alumina and Polymers, e.g. Nylon) and with different deposition techniques such as filtration. The membrane exhibits extremely high permeabilities up to ~40.000 barrer (H₂) and moderate selectivity. The presented material is very interesting.

However, the manuscript structure is a bit confusing and I see several points necessary to make the paper suitable for publication in Nat. Commun.

Reply: Thanks a lot for the positive comments. Here we would like to reply to your comments:

Some minor remarks:

Line 159 – “1D nanotubes” – since you have multilayers (thickness of at least 5nm) of your material your nanotubes correspond to 3D structure. I am not convinced that it is only a monolayer. I would simply refer to “nanotubes”, without dimensions.

Line 196 – “concerntation” should be concentration

Line 281 – “selevtivity” should be selectivity

You should read the paper carefully again to ensure there are no typos anymore.

Reply: Thanks for pointing it out. Apart from correcting these two errors, we checked the entire manuscript again.

Action: Line 159: “1D nanotubes” changed to “nanotubes”

Line 196: “concerntation” changed to “concentration”

Line 281: “selevtivity” changed to “selectivity”

Some major remarks:

1) Page 8, 171 – you have such a narrow size distribution, that you should be able to see it on the XRD, since it must be ordered in some way. XRD data should be added to your general characterization methods? Compare for PIMs: <https://doi.org/10.1002/adsu.201800044>

Reply: Thanks. In the abovementioned paper <https://doi.org/10.1002/adsu.201800044>, the XRD profiles indicate the presence of chain arrangements within the membrane induced by van der Waals forces. This is normal for certain 1D polymers. However, our CMT is a 3D cross-linked porous polymer constructed *via* irreversible C–C coupling bonds, which normally lead to amorphous porous polymers. Supporting figure 15 shows the XRD profile of CMT, which has no obvious peak.

In fact, 2D or 3D crystalline porous polymers (also known as covalent organic frameworks (COFs)) are normally constructed with reversible bonds, such as B–O,

C–N, B–N, and B–O–Si. Preparing crystalline COFs with irreversible bonds, such as C–C coupling bonds, is a big challenge in polymer chemistry (O. M. Yaghi, *Chemistry of Covalent Organic Frameworks. Acc. Chem. Res.* 48, 3053–3063 (2015). “*The future challenges pertain to extending this covalent chemistry to C–O, C–C, and other such strong bonds, which will provide access to a whole new area of useful materials not the least of which are new forms of carbon.*”).

Comparing to conjugated microporous polymers synthesized in solvents with noble metal catalysts under stirred conditions, our CMT shows a remarkably narrow pore size distribution. We believe that our endogenous liquid-state polymerization (that is, without the presence of solvents, initiators, or catalysts) is beneficial for achieving a homogeneous microporous structure, because precursor molecules are polymerized in the isotropic liquid state, without randomization effects of solvents and catalysts.

Supporting figure 1. XRD profile of CMT

Action: XRD profile of CMT is included in the Supplementary Information as Supplementary figure 15.

2) In Figure 1e there is a nice AFM picture of the single sheet. In Figure 2h there is another AFM image. The latter is not really needed to state your points, everyone would agree that your sheet thickness stays the same after filtration.

Reply: Thanks for pointing it out.

Action: The AFM image in Figure 2 is removed.

3) In Figure 2e you show your membrane prepared by filtration, with the SEM top view and cross section in Figure 2 f/g. This membrane is 20 μm thick and was not used for any of your experiments, as far as I see it. Why do you not give the actual membrane, that was used for your experiments in the main section (which now is in SI Fig. 16). I cannot really understand your filtration technique to prepare a

membrane, when you afterwards measure a different membrane, which is directly synthesized.

Reply: Thanks. In figure 2, we are demonstrating the solution-processability of CMT nanosheets, which enables us to obtain a free-standing membrane. This is a big contrast to reported conjugated microporous polymers, which are normally synthesized as fine powders with poor processability (*Angew. Chem., Int. Ed.* 48, 9457 (2009). *Adv. Mater.* 23, 3723 (2011).). Figure 2g is the SEM image of the cross-section of a free-standing membrane with a thickness of ~20 μm .

In our gas separation measurements, we actually tested CMT membranes with four thicknesses (0.5, 1, 5 and 13 μm). (The gas data of the four membranes were presented in Figure 3a in our original manuscript. Now, the data is moved to supplementary information as Supplementary figure 16 in the revised manuscript.) To avoid repeatedly presenting cross-section images, we chose not to present again the cross-sections of CMT membranes we used for gas separation measurements.

Action: CMT membranes with different thicknesses (0.5, 1, 5 and 13 μm) were prepared for gas separation measurements. The data is presented as Supplementary figure 16.

4) It is completely unclear, where your data originates from. You write that you use a 1 μm thick membrane for H₂/CO₂ mixed gas in line 252-253. Then, in line 257-259 you speak of a 0.5 μm thick membrane, where you collected your single gas permeation data on.

Keep it consistent and measure it all for the same sample (and add error bars from your control experiments). And if you measure it again, make sure to measure everything at elevated temperatures and in mixed gas permeation (see point 5).

Reply: Thanks for pointing it out.

We used a 1 μm thick membrane for H₂/CO₂ mixed gas measurements.

Four membranes (thickness in 0.5, 1, 5 and 13 μm , respectively) were used for single gas measurements and they all showed similar performances (the gas data of the four membranes were presented in Figure 3a in our original manuscript). We apologize for not making this clearer in the previous version of the manuscript.

Action: To avoid confusion, only the gas data with error bar of the 1 μm thick membrane we used for single gas measurement is presented in Figure 3a, and is referred to in Line 255-260 of the revised manuscript. The data of membranes with different thickness is moved to the Supplementary Information as Supplementary figure 16).

5) The next thing I find is Figure 3 – You compare the Robeson plot with your ideal permeability and selectivity - this is not consistent. You should easily use your “real” permeability data from mixed gas measurements of the binary mixtures. I think you would still be better than all the other points. However, there is often a strong variation between ideal and real permeation data. You should also check back on your data points, whereas the data points originate from real or ideal permeation data and if it is comparable.

Reply: Thanks. As a new material, our CMT possesses permanent pores with extreme structural rigidity similar with inorganic microporous solids, such as zeolites and MOFs. At the same time, CMT can be solution-processed into membranes, similar to linear polymers, such as PIMs and PTMSP. Thus, its gas separation performances are compared with inorganic microporous solids and solution-processable linear polymers. For the real permeation data, testing conditions, like temperature or transmembrane pressure, could be largely varied for inorganic microporous solids and linear polymers. Thus, we used ideal permeation data for all the materials for comparison in Robeson plots. In fact, the Robeson upper bound plots are intended for pure gas data, although mixed gas data is sometimes plotted. In the revised manuscript, we have also added the real (mixed gas) permeation data in Figure 3b.

Action: We confirmed that all the data in Robeson plots Figure 3b, 3c and 3d are ideal permeation data. The real permeation data is also added in Figure 3b.

6) The same point as in my comment (5) – Line 281-282.

Action: We confirmed that all the data in Robeson plots Figure 3b, 3c and 3d are ideal permeation data.

7) Line 310-313. Your data seems to be inconsistent here. When heating up neat materials outgoing from general physical knowledge about adsorption and diffusion, even in zeolites, MOFs, COFs and especially in polymers (solution diffusion), your permeability ALWAYS increases, whereas your selectivity ALWAYS decreases. This is the general trade-off in membrane science. I can also not find the data set on this study in the supporting information, where you measure at different temperatures.

Reply: Thanks for pointing it out. We agree with the reviewer that in general, most gas permeability increases with temperatures, and selectivity decreases. There are sometimes deviations, although they occur mostly in the case of CO₂. Since gas permeability is defined as solubility × diffusivity, CO₂ permeability for some membranes showing high affinity to CO₂ may decrease at higher temperature, when solubility decreases. Therefore, the H₂ permeability is increased together with the increase of H₂/CO₂ selectivity. Such phenomena have been observed for PBI membranes (*Journal of Membrane Science*, 2011, 375, 231), cross-linked P84 membranes (*Journal of Membrane Science*, 2011, 572, 118), Amine-functionalized (AI) MIL-53/VTEC™ mixed-matrix membranes (*Journal of Membrane Science*,

2017, 530, 201) and 6FDA-Durene/ZIF membranes (*Advanced Materials*, 2017, 29, 1603833). For our membrane, a higher testing temperature at 150 °C causes the membranes to have a broader free volume that facilitates a higher H₂ permeability. Meanwhile, as shown in the CO₂ sorption isotherms at different temperatures (Supplementary figure 18b), the concentration of adsorbed CO₂ inside the membranes decreases as a function of increasing temperature owing to the larger thermal energy of CO₂ at higher temperatures. Due to the decrease of CO₂ adsorbed concentration, the solubility coefficient decreased ($S=C/P$, C refers to the concentration of adsorbed CO₂, P refers to the applied pressure), thereby reducing the permeability of CO₂. Therefore, the H₂ permeability is increased together with the increase of H₂/CO₂ selectivity for our membrane. In our manuscript, the long-term gas separation performance at high temperature is mainly used for demonstrating its structural stability, and the temperature dependence is not a key component.

Action: We have provided the data and an explanation for this observation with CO₂ sorption data at Supplementary figure 18.

“It shows that the concentration of adsorbed CO₂ inside the membranes decreases as a function of increasing temperature owing to the larger thermal energy of CO₂ at higher temperatures. Due to the decrease of CO₂ adsorbed concentration, the solubility coefficient decreased ($S=C/P$, C refers to the concentration of adsorbed CO₂, P refers to the applied pressure), thereby reducing the permeability of CO₂. Therefore, the H₂ permeability is increased together with the increase of H₂/CO₂ selectivity.”

After the manuscript is subjected to a major revision considering the points listed above, it may be suitable for a publication in Nat. Comm.

REVIEWERS' COMMENTS:

Reviewer #2 (Remarks to the Author):

I am happy with the replies to my comments. Paper is now suitable for publication.